# A systematic review of the untreated mortality of murine typhus

**Johannes F. Doppler**[1,2]*, **Paul N. Newton**[1,2,3]

**1** Lao-Oxford-Mahosot Hospital-Wellcome Trust Research Unit (LOMWRU), Microbiology Laboratory, Mahosot Hospital, Vientiane, Lao PDR, **2** Faculty of Infectious and Tropical Diseases, London School of Hygiene and Tropical Medicine, London, United Kingdom, **3** Centre for Tropical Medicine and Global Health, Nuffield Department of Medicine, Old Road Campus, University of Oxford, Oxford, England

* j.doppler@gmail.com

**Data Availability Statement:** All relevant data are within the manuscript and its Supporting Information files.

**Funding:** The authors received no specific funding for this work.

## Abstract

Murine typhus is an acute febrile, flea-borne disease caused by the bacteria *Rickettsia typhi*. The disease occurs worldwide but is likely underrecognized due to its non-specific symptoms, causing significant morbidity. A systematic review found disease complications in one-fourth of all patients and a long fever duration in those untreated. Although mortality in treated cases is estimated to be very low, some case series have shown a notably higher mortality in untreated patients. This study aimed to describe the outcomes and estimate the mortality of untreated murine typhus through a comprehensive systematic literature review. We systematically searched the literature for articles describing untreated murine typhus patients, excluding cases with no laboratory assay confirmed diagnosis, those who received efficacious treatment, had incomplete information on primary outcome and articles describing less than 10 patients and performed a narrative synthesis of the study findings. The study protocol followed the PRISMA guidelines and was part of a more extensive protocol registered at PROSPERO (CRD42018101991). Twelve studies including a total of 239 untreated patients matched the eligibility criteria. Only a single study reported one death in 28 patients, giving a patient series mortality of 3.6% and an overall mortality of 0.4% in 239 untreated patients. Complications were reported in 10 of the 12 studies and included involvement of the central nervous system, kidney and lung, with a hospitalisation rate of 70% and ICU admission rate of 27% in one study. The mean duration of fever in untreated patients was 15 days in two and 12.7 days in one study. Although the untreated mortality in this study was low, the sample size was small. Murine typhus caused significant morbidity when untreated, leading to high hospitalisation rates and highlighting the importance of early diagnosis and treatment of this neglected disease to reduce disease burden and health-care related costs.

## Author summary

Murine typhus is an acute febrile, flea-borne bacterial disease that has been reported worldwide and continues to cause significant morbidity when untreated. The often self-

**Competing interests:** The authors have declared that no competing interests exist.

limiting, non-specific clinical symptoms of the disease resemble that of common viral illnesses, suggesting that the disease is underdiagnosed. While the mortality in treated cases is estimated to be very low, disease complications in one-fourth of all patients and a prolonged duration of fever in untreated cases have been reported. We systematically searched the literature to identify articles describing laboratory diagnostically confirmed clinical cases of untreated murine typhus and summarized disease outcomes, including mortality, of patients in eligible studies. Of the 12 studies containing 239 untreated patients that matched the eligibility criteria, only one study reported a single death amongst 28 untreated patients, resulting in a patient series fatality rate of 3.6% and an overall untreated fatality rate of 0.4%. Disease complications were mentioned in 10 of 12 studies and the mean duration of fever in untreated cases was 15 days in two studies and 12.7 days in one study, demonstrating the significant morbidity caused by untreated murine typhus and highlighting the importance of early diagnosis and treatment of this neglected disease.

## Introduction

Murine typhus is an acute febrile illness caused by the bacteria *Rickettsia typhi*. The disease, transmitted to humans by fleas through rodent reservoirs, has a worldwide occurrence, with patients reported from varying environments and regions including Southeast Asia, North America, the Mediterranean and Northern Africa [1–3].

It is likely a highly underrecognized disease due to its usually self-limiting, non-specific clinical symptoms that may resemble those of common viral illnesses. [2].

Murine typhus is usually considered to be a mild illness with a very low fatality rate in treated patients, with fever receding in a mean of approximately 3 days with adequate antibiotic treatment, usually with doxycycline [4, 5]. However, fever duration between 12 to 21 days in those untreated with an overall complication rate of 26% (including pulmonary, central nervous system involvement and acute kidney injury) have been reported [3], demonstrating the significant burden caused by this pathogen.

This burden not only translates to loss of productivity due to long illness duration but potentially also leads to increased health-care costs until a diagnosis and treatment of murine typhus is established [6].

Estimates of an untreated mortality of approximately 4% in murine typhus patients can be found in the literature [2]. This probably represents an overestimation since differentiation of *R. typhi* from *Rickettsia prowazekii* is difficult. *R. prowazekii*, the agent of epidemic typhus, also belongs to the typhus group rickettsiae and causes similar symptoms but usually has a significantly higher mortality. Historic diagnostic tests that were commonly used in the pre-treatment era, such as the Weil-Felix, lack specificity and even in contemporary serological assays human antibodies against *R. typhi* cross-react extensively with *R. prowazekii* antigen and vice versa [7]. This systematic literature review estimates the mortality and describes outcomes of laboratory assay confirmed cases of untreated murine typhus.

## Methods

The study protocol followed the PRISMA guidelines (S1 PRISMA checklist) and was part of a more extensive protocol that has been registered at PROSPERO (CRD42018101991).

### Eligibility criteria

Studies reporting laboratory assay diagnostically confirmed cases of symptomatic murine typhus infection in humans were eligible for inclusion. Criteria for exclusion of studies was usage of the Weil-Felix reaction as the only diagnostic test because of its low accuracy at differentiation of *R. typhi* from *R. prowazekii*, treatment of patients with medicines known to be efficacious or to have some efficacy (the tetracyclines, chloramphenicol, fluoroquinolones, para-aminobenzoic acid), incomplete information on diagnosis, treatment or primary outcome, reporting of less than five untreated patients or less than 10 patients including those treated to reduce selection bias. There were no restrictions on publication period, study language or type of study design. Records describing rodents and vectors were not included.

### Information sources, search strategy and study selection

Global Health (1910 to present), Embase Classic/Embase (1947 to present) and Medline (1946 to present) were searched individually on June 28th, 2018 using a search strategy that included all synonyms and database-specific subject headings of murine typhus (S1 Fig). One author (JFD) deduplicated the search results and screened the titles and abstracts of the retrieved studies for eligibility. Full-texts of articles were obtained for studies potentially matching the eligibility criteria and in case a decision on eligibility could not be made from screening the title and abstract. If doubts remained after reviewing the full-text consensus was sought with the other author. Articles in English, French, Italian, Spanish, Portuguese and German were reviewed in original language, articles in other languages were translated using Google translate. Identical patient series encountered in different papers were identified and excluded.

### Data extraction, bias assessment, outcome and summary measures

Using a standardised data extraction form, data on year and region of study, study design, number of untreated cases, demographic and clinical characteristics, diagnostic test used, complications, number of patients hospitalised and number of deaths in untreated cases were extracted from eligible studies. Included studies were assessed for bias by grading each study by patient selection, diagnostic test and patient information, using a form adapted from a published systematic review using similar methodology [8] (S1 Table).

Primary outcome was mortality in untreated murine typhus patient series, secondary outcomes were days of fever, clinical features and complications. Because of the heterogeneity of the study data, a narrative synthesis was conducted.

## Results

### Study selection

Screening of search results identified 148 potentially eligible articles for which full-texts were sought. Of these, 134 articles did not match the eligibility criteria after reviewing the full-text and for further two articles full-text was not retrievable, which resulted in a total of 12 studies included for analysis. A summary of the study selection process is given in Fig 1, for a list of articles excluded after full-text review see S2 Table.

### Study characteristics

The 12 included studies contained a total of 239 untreated patients, with the smallest series containing five and the largest 60 untreated patients. The series cover a time span from 1942 to 2016. Three studies were conducted in the years before efficacious antibiotic treatment was available, which includes the only study with a reported death attributed to untreated murine

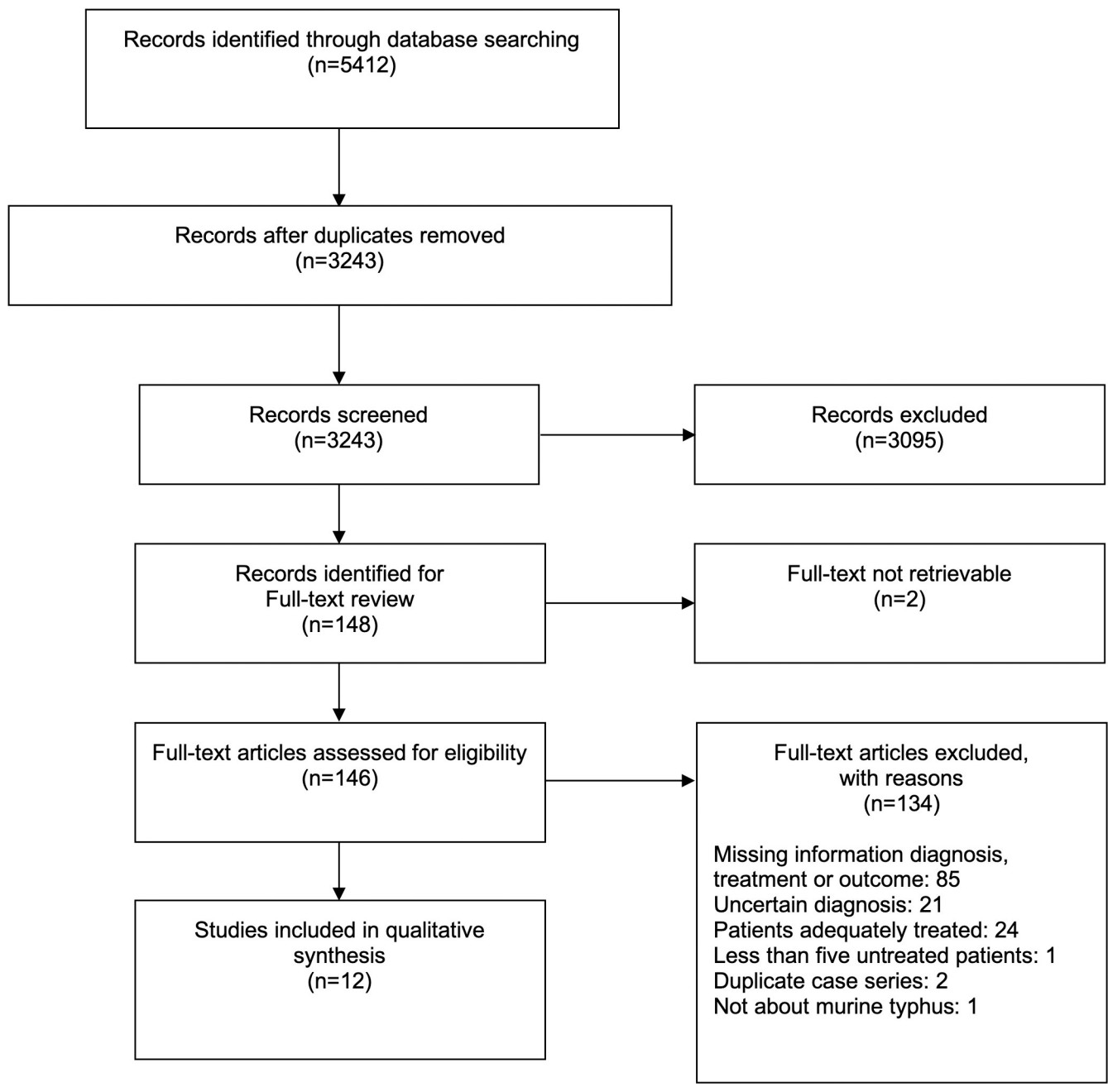

**Fig 1. Study selection process flow diagram.**

typhus [9]. This study retrospectively reviewed patients in the British Army Middle East Forces. The other included studies were a prospective study of an outbreak in Panama in 1947 [10], a trial of drug therapy with para-aminobenzoic acid compared with no treatment in Puerto Rico [11], a retrospective chart review of patients presenting at a hospital in Israel from 1976 to 1985 [12], another reviewing patients at four hospitals in southern Texas from 1980 to 1987 [13], a prospective and retrospective study of patients in a hospital in Spain from 1979 to 1995 [14], a second study from Spain reviewing patients presenting at a hospital in the Canary islands from 2000 to 2002 [15], a prospective study of Bedouin children presenting at a hospital

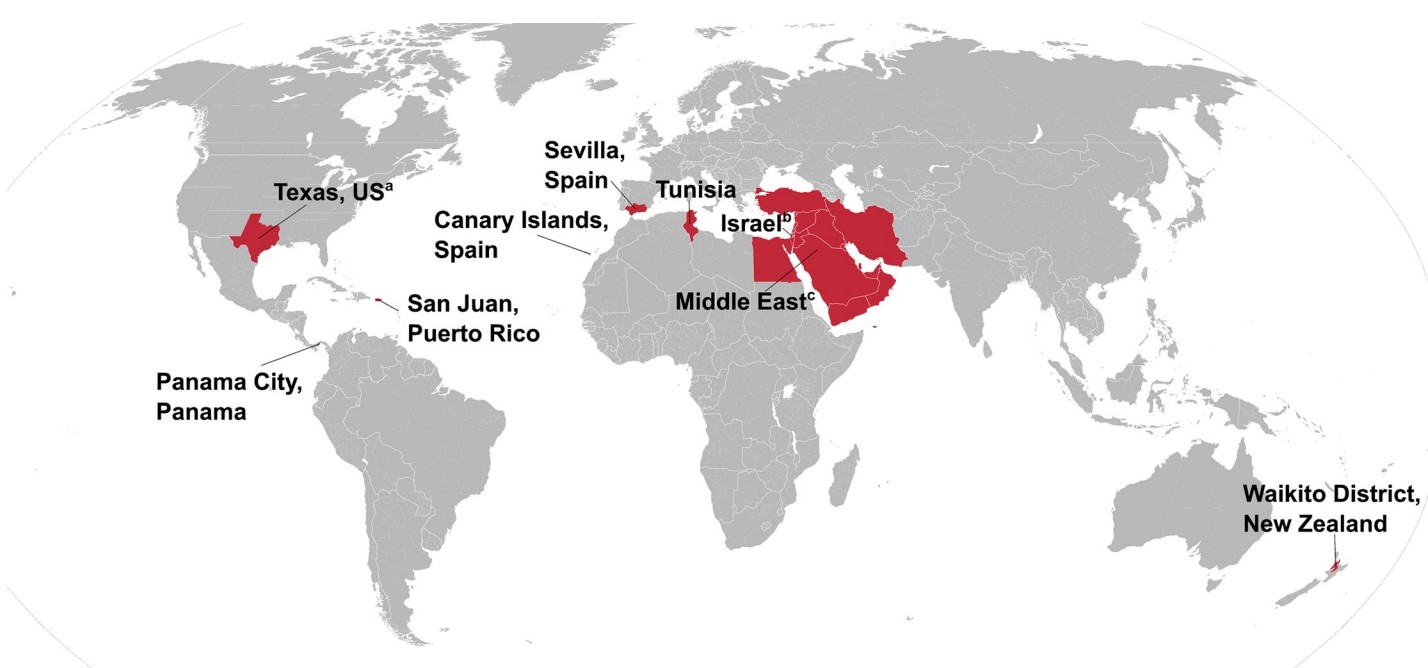

**Fig 2. Geographic distribution of included studies describing the untreated mortality of murine typhus.** [a] Three studies. [b] Two studies. [c] Study in the Middle East Forces, countries not specified. Map from https://en.wikipedia.org/wiki/File:Blank_Map_World_Secondary_Political_Divisions.svg, modified using Inkscape v1.0.

in Israel from 2003 to 2005 [16], a retrospective study of patients diagnosed in a district in New Zealand in 2006 [17], a retrospective outbreak investigation in Texas conducted by the CDC and Texas authorities in 2008 [18], a retrospective study of patients diagnosed in Tunisia from 2006 to 2008 [19] and a retrospective chart review of patients presenting at two hospitals in Texas from 2013 to 2016 [20]. All included studies were published in English.

Apart from the studies in New Zealand, Tunisia and the outbreak investigation in Texas all studies were hospital-based. Treated patients were encountered in 83% (10/12) of the included studies and these were excluded from the untreated patient series. The geographic distribution of the included studies is shown in Fig 2 and a summary of the characteristics of the included studies is given in Table 1. Demographic and clinical characteristics were often only available for the entire study population, and this information was provided in the table when there was no specific information available for the untreated patient group. In summary, for the untreated group, information on age and sex was available for 17% (2/12) and on fever duration and complications for 33% (4/12) of the studies.

## Risk of bias within studies

The risk of bias within studies was assessed using a standardised form (see Methods). Of the 12 included studies, three were prospective, seven retrospective, one pro- and retrospective and for one study it was not specified. In 10 of 12 studies patients had to be excluded because of efficacious treatment, which resulted in the exclusion of 367 patients. Only one of the 12 included studies specified measuring *R. prowazekii* titers for all patients to differentiate endemic from epidemic typhus. Ten studies did not differentiate from *R. prowazekii* and in one study three of 13 patients were diagnosed only using the Weil-Felix test. There was a high frequency of missing clinical and patient information across the included patient series, only one study had no missing information. Details of the bias assessment for each study can be found in S3 Table.

**Table 1. Characteristics of included studies, sorted by year of study.**

| First author, Journal [Reference] | Region, Study period | Study design, characteristics | No. untreated patients | Age, Years (mean) | Sex male (%) | Duration of fever, days | Diagnostic test used | Complications | Patients hospitalised (%) | Untreated deaths % (n) |
|---|---|---|---|---|---|---|---|---|---|---|
| **Crofton, JRAMC [9]** | Middle East, countries not specified 1942–1943 | Retrospective case series Middle East Forces | 28 (28/28) | - | - | - | Rickettsial agglutination test with differentiation from R. *prowazekii* for all patients plus Weil-Felix. | Five "severe" cases, of which one died (post-mortem—lung with haemorrhagic bronchopneumonia and abscess formation, brain with microscopic haemorrhages and scanty cellular nodes) | all | 3.6% (1/28) |
| **Calero, AJTMH [10]** | Panama City, Panama 1947 | Prospective case series Report of an outbreak | 13 (13/13) | 16–43 | 11/13 (85%) | 12–17 (15) | Ten cases confirmed by complement or rickettsial agglutination test with differentiation from *R. prowazekii* plus Weil-Felix, three cases by Weil-Felix only | Seven patients hospitalised Pronounced "stuporous condition" in 30.7% | 7/13 (54%) | 0% (0/13) |
| **Diaz-Rivera, Am J Med Sci [11]** | San Juan, Puerto Rico 1947–1948? | Prospective case series Drug trial, 33 treated | 27 (27/60) | 15–60 (30.6) | 18/27 (67%) | 12–22 (15) | Complement fixation test with conversion from negative to positive plus Weil-Felix with increasing titers up to ≥ 1:160 | - | all | 0% (0/27) |
| **Shaked, Infection [12]** | Israel 1976–1985 | Retrospective case series Chart review | 11 (11/45) | - | - | 12–19 | Positive complement fixation test or immunofluorescent antibody test for murine typhus with differentiation from spotted fever (unspecified antigens/titer) | One case of encephalitis Two cases of pneumonia and one scrotal ulcer in treated patients | all | 0% (0/11) |
| | | 34 treated | | - | 34/45[a] (76%) | | | | | |
| **Dumler, JAMA [13]** | Southern Texas, US 1980–1987 | Retrospective case series Chart review | 8 (8/80) | - | - | - | Indirect fluorescent antibody test (IFA) with 4-fold rise in typhus group antibody titer, single high titer ≥ 1:128 and lower titer to spotted fever group (*R. rickettsii)* antigen than to typhus group antigen No differentiation from *R. prowazekii*, previous studies revealed no cases of *R. prowazekii* infection | No complications in the untreated Neuropsychiatric complications (confusion, stupor, coma, hallucinations) in six, seizure in three, ataxia in one, renal insufficiency in five, jaundice in two, respiratory failure in three and hematemesis in one of the treated patients Seven patients admitted to ICU, three (3.8%) died. Two deaths due to shock, renal and multisystem failure | 77/80 (96%) | 0% (0/8) |
| | | 72 treated | | - (46.3)[a] | - (40%)[a] | - | | | | |

(*Continued*)

**Table 1.** (Continued)

| First author, Journal [Reference] | Region, Study period | Study design, characteristics | No. untreated patients | Age, Years (mean) | Sex male (%) | Duration of fever, days | Diagnostic test used | Complications | Patients hospitalised (%) | Untreated deaths % (n) |
|---|---|---|---|---|---|---|---|---|---|---|
| **Bernabeu-Wittel, Arch Intern Med [14]** | Sevilla, Spain 1979–1995 | Prospective case series 1983–1995 62 cases Retrospective case series 1979–1982 42 cases Hospital in Spain | 60 (60/104) | - | - | - (12.7) | Immunofluorescence antibody assay *R. typhi* IgG ≥ 1:512 in 76 patients, 4-fold rise in titer in 28 patients, differentiation from *Rickettsia conorii* | Organ complications in four untreated patients (6.7%) Nine of all 104 patients (8.6%) with organ complications, including pneumonitis in six, cerebellitis in one and multiorgan failure in two patients Complications classified as severe in four of 104 patients (3.8%) | all | 0% (0/60) |
| | | 44 treated | | 12–81[a] (37.9) | 57/ 104[a] (55%) | 8–27[a] (12.5) | | | | |
| **Hernández-Cabrera, Emerg Infect Dis [15]** | Gran Canaria, Canary Islands, Spain 2000–2002 | Not specified in- and outpatients (>14 years) at University Hospital Las Palmas | 8 (8/22) | - | - | - | Direct immunofluorescence antibody test *R. typhi* IgM ≥ 1:40 or 4-fold rise in IgG titer and differentiation from *Rickettsia conorii* | No complications in the untreated One renopulmonary syndrome, one encephalitis and one meningitis with renal failure in the treated | Not specified | 0% (0/8) |
| | | 14 treated | | 14–76[a] (28) | 21/22[a] (95%) | 7–20[a] (10) | | | | |
| **Shalev, Scand J Infect Dis [16]** | Rahat, Israel 2003–2005 | Prospective case series Bedouin children | 47 (47/76) | - | - | - | Micro immunofluorescence *R. typhi* titer IgM or IgG ≥ 1:100 in acute and/or convalescent sample and stronger reaction to *R. typhi* antigen than to spotted fever group (*R. conorii*) antigen | No complications | 1/76 (1.3%) | 0% (0/47) |
| | | 29 treated | | - (7.3)[a] | 39/76[a] (51%) | - | | | | |
| **Gray, N Z Med J [17]** | Waikato region, New Zealand 2006 | Retrospective case series | 5 (5/12) | - | - | - | IFA with 4-fold *R. typhi* antibody titer rise in six patients and single high IgM ≥ 1:512 or IgG ≥ 1:1024 in six patients | - Nine patients hospitalised | 9/12 (75%) | 0% (0/5) |
| | | Seven patients treated | | 19–69[a] (46) | 6/12[a] (50%) | - | | | | |
| **Adjemian, Emerg Infect Dis [18]** | Austin, Texas, US 2008 | Retrospective case series Outbreak investigation | 16 (16/33) | - | - | - | IFA with 4-fold antibody titer rise to *R. typhi* in all and detection of DNA in clinical specimen by PCR in one patient | - 23 of all 33 patients hospitalised and 9 of them admitted to the ICU (pneumonia, coagulopathy, renal failure) | 23/33 (70%) | 0% (0/16) |
| | | 17 treated | | 7–64[a] (39) | - (56%)[a] | - | | | | |
| **Znazen, Med Mal Infect [19]** | Tunisia 2006–2008 | Retrospective case series | 8 (8/43) | - | - | - | Micro immunofluorescence assay with IgM titer ≥ 1:32 to *R. typhi* in 32 and seroconversion or significant titer rise in 11 patients, sera showing cross-reactions between *R. typhi* and *R. conorii* or *R. felis* were excluded | - 36 patients hospitalised | 36/43 (84%) | 0% (0/8) |
| | | 35 patients treated | | 8–83[a] (41) | - (51%)[a] | - | | | | |

*(Continued)*

**Table 1.** (Continued)

| First author, Journal [Reference] | Region, Study period | Study design, characteristics | No. untreated patients | Age, Years (mean) | Sex male (%) | Duration of fever, days | Diagnostic test used | Complications | Patients hospitalised (%) | Untreated deaths % (n) |
|---|---|---|---|---|---|---|---|---|---|---|
| **Afzal, Emerg Infect Dis [20]** | Hidalgo County, Texas, US 2013–2016 | Retrospective case series Review of records from two hospitals | 8 (8/90) | - | - | - | IFA with IgM or IgG titer ≥ 1:128 to typhus group rickettsiae in 87, 4-fold IgG titer rise in three patients | - Complications in 25 patients, including bronchiolitis in two, pneumonia in eight, pancreatitis in three, cholecystitis in one, myositis in one, rhabdomyolysis in two, meningitis in two, sepsis with acute kidney injury in one, septic shock in four patients and septic shock plus pneumonia in one patient. 13 patients required ICU care | all | 0% (0/8) |
| | | 82 patients treated | | - | 45/90[a] (50%) | - | | | | |

"-"refers to missing information.

[a] entire study population, including treated patients.

## Primary outcome

The primary outcome was available for all included studies. The only death among all untreated patients diagnosed with murine typhus occurred in the study of 28 patients in the Middle East Forces, resulting in a fatality rate of 3.6% for this patient series. The patient died on the 16th or 17th day of disease and the post-mortem study showed a haemorrhagic bronchopneumonia with abscess formation in the lung and microscopic haemorrhages with scanty cellular nodes in the brain.

When combined with all other patient series that reported no deaths, there was one death in 239 untreated patients, corresponding to a fatality rate of 0.4%.

## Demographics and secondary outcomes

Specific information on age and sex was available only for 17% (40/239) of the untreated patients. In this group, the age range was 15–60 years, with a mean age of 30.6 years specified in one study, and the male percentage was 73% (29/40). When combined with findings from studies that reported demographic information inclusive of treated patients, 59% (231/389) of diagnosed patients with murine typhus were male. The study among the Middle East forces did not give demographic information but likely further skewed the sex distribution. Only one study reported an excess of female patients.

## Duration of fever

In the studies that provided information on the duration of fever in untreated patients, the minimum specified duration of fever was 12 days, with the maximum ranging from 17 to 22 days. The mean duration of fever was 15 days in two studies while one study reported a mean of 12.7 days. This latter study noted a mean duration of fever of 12.5 days when treated patients

**Table 2. Key outcome findings.**

| Outcome | Findings in patients with untreated murine typhus |
|---|---|
| *Days of fever* | Range 12–22 days in 51 patients<br>Mean of 15 days in 40 patients, 12.7 days in 60 patients |
| *Complications* | No complications (0%) among 47 Bedouin children<br>Five (18%) "severe cases" (including the death) among 28 patients in the Middle East Forces<br>Seven (54%) of 13 patients hospitalised in an outbreak in Panama, pronounced "stuporous condition" in 30.7%<br>One (9%) case of encephalitis among 11 patients in a hospital in Israel<br>Four (6.7%) of 60 patients with organ complications in a hospital in Spain |
| *Mortality* | 3.6% (1 death) among 28 patients of the Middle East Forces<br>0.4% for the entire patient series (239 patients) |

were included. Specific information on fever duration in untreated patients was available in 46% (111/239).

## Complications

Table 2 gives an overview of the key outcome findings in untreated patients of the included studies. Of the seven studies which examined complications in untreated patients, only the study conducted among Bedouin children recorded no complications. Two studies reported no complications in the untreated, but described significant complications in the treated patients: in one study three of 14 treated patients had renopulmonary or central nervous system (CNS) complications and in the other study complications included renal, pulmonary, gastrointestinal and CNS involvement, with seven of 72 treated patients admitted to an intensive care unit and three deaths due to shock, renal and multisystem failure. Of the studies that specified complications in a cohort inclusive of treated patients, three studies noted hospitalisation rates of 84% (36/43), 75% (9/12) and 70% (23/33), with nine of the 33 patients (27%) admitted to an intensive care unit because of pneumonia, coagulopathy or renal failure, one study stated complications including pneumonitis, cerebellitis and multiorgan failure in nine (8.6%) out of 104 patients with murine typhus, in four (3.8%) of those classified as severe and one study reported complications in 25 of 90 patients (28%) including pulmonary, CNS, gastrointestinal and muscular involvement, with 13 of 90 (14%) requiring ICU care.

Five of the 12 studies had all patients hospitalised, more than 50% of the patients were hospitalised in a further five studies, 1.3% in one study and in one study it was not specified.

## Discussion

This review of 239 untreated murine typhus patients found a low fatality rate of 0.4% but reported significant morbidity, including a mean fever duration of up to 15 days, hospitalisation rates above 50% and complications in 6–30% with pulmonary, renal, gastrointestinal, muscular and CNS involvement. No complications were found in a study among 47 Bedouin children [16], potentially indicating a possibly milder course in the younger age group. Similarly, another study not eligible for inclusion in this review found a usually mild to moderate systemic illness but rarely severe complications in 97 children in Texas [21].

Information on demographics and secondary outcomes for untreated patients was limited since 10 of the 12 studies also contained treated patients and information was often only available for the entire study population. The review suggested a predominance of male patients and only one study reported an excess of female patients.

Nine of the 12 included studies were from a period in which antibiotic treatment was widely used and all of them included treated patients. None of these were randomised clinical trials,

and in the antibiotic era only patients with a mild and self-limiting course remained untreated while patients with more severe disease received antibiotics. Additionally, reporting of severe complications or deaths in untreated patients in these antibiotic era studies would have caused ethical concerns. Since today efficacious treatment is often available, it is therefore unlikely that complications and fatalities in untreated murine typhus patients will be reported in the literature, apart from in retrospective case series. Two studies in this review reported no complications in the untreated, but significant complications in the treated patients including three deaths. This may be because those with severe disease had a higher probability of receiving antibiotics.

This study tried to limit the possibility of including confounding *R. prowazekii* infections in the studies reviewed by only including laboratory assay confirmed patient series of murine typhus. However, the exclusion of studies that used the Weil-Felix test as the only diagnostic method significantly restricted the number of eligible studies. The Weil-Felix test was the major diagnostic test in the decades before efficacious antibiotic treatment was available and a substantial number of studies from this period reporting deaths from untreated murine typhus had to be excluded–whether they represented epidemic or murine typhus is unknown. The Weil-Felix test is still commonly used in low resource settings today, where infrastructure for modern laboratory assay confirmed diagnosis is not available. This setting is especially encountered in poorer countries from which studies are expected to be underrepresented in this review. Furthermore, the requirement of a laboratory assay confirmed diagnosis likely led to a tendency of selecting studies that were hospital-based, in which the necessary infrastructure was available and where patients likely received better care. Alternatively, only patients with a more severe disease course might have sought hospital care. In this review nine of the 12 included studies were hospital-based.

But even with contemporary serological assays, differentiation of *R. typhi* from *R. prowazekii*, which has an estimated mortality of ~20% in untreated patients [22], is difficult. Human antibody response to *R. typhi* antigen cross-reacts with *R. prowazekii* antigen and while the two diseases can sometimes be distinguished by comparing the titers against *R. typhi* and *R. prowazekii*, a definite differentiation often requires additional techniques such as Western blotting and cross-adsorption, which is costly and restricted to laboratories with facilities for safe culturing of rickettsiae [7]. Only one of the 12 included studies specified measuring *R. prowazekii* titers for all patients to differentiate from epidemic typhus and no study reported using additional techniques for differentiation, therefore it cannot be excluded that some cases in this review might have been due to *R. prowazekii* infection.

The first more widely used serological test with better accuracy at identifying infection with *R. typhi* was introduced in 1942 [23], but widespread use of effective antibiotic therapy started soon after its discovery in the beginning of the nineteen-fifties [24]. This leaves a small window of a few years when diagnostic methods were available to exclude epidemic typhus but effective treatment was still unknown. Unfortunately, for this period, few studies were encountered that described patient series of untreated murine typhus with outcome measures, and later studies had to deal with the ethical issue of reporting untreated patients in an era when effective antibiotic treatment was available. Although a better powered estimate of the untreated mortality of murine typhus could be obtained from historic records that used the Weil-Felix test for diagnosis, this would require subjective linking of test results with clinical, epidemiological and environmental information, including knowledge of local historic occurrence of *R. prowazekii*. However, especially in older studies, there might be confusion with Brill-Zinsser disease, a recrudescent form of epidemic typhus with usually milder symptoms that may resemble those of murine typhus.

Fatality rates are likely to be influenced by multiple factors, including clinical syndrome, age, nutritional status and intensity of supportive care but our sample size does not allow such analysis. In one study the hospital mortality of *R. typhi/Rickettsia* spp. central nervous system infections was estimated at 18% [25].

The estimated untreated mortality of murine typhus in our review is low when compared with other findings in the literature. Civen et al. stated a fatality rate of 4% in untreated patients and 1% in patients with antibiotic treatment [2]. In one of the included studies in our review, the fatality rate among treated, hospitalised murine typhus patients was 3.8% [13] while in another study, not included here, a fatality rate of 0.4% was reported in a patient series of 3,048 murine typhus cases from 1985 to 2015 in Texas, a time period in which effective antibiotic treatment was available [26]. Possible explanations for these differences in reported fatality rates might be differences in patient demographics and clinical features, including age and comorbidities, differences in time from symptom onset until effective treatment is given, variability in pathogen virulence and a bias towards more severity in hospital-based studies.

## Conclusion

This is the first systematic review, as far as we are aware, to estimate the mortality in untreated murine typhus patients. Considering the small number of included studies, small overall sample size and the high proportion of studies conducted in times of efficacious treatment, the estimates of morbidity and mortality in untreated patients reported in this review might be of low accuracy. A larger sample size and more studies conducted in different settings would be necessary to reach a more accurate estimate but with treatments available this would be unethical. We conducted a review of the untreated mortality of scrub typhus (*Orientia tsutsugamushi*), for which significantly more data are available (76 studies, 19,644 patients), giving an estimated median (range) untreated mortality of 6.0 (0–70) % [7], higher than we describe for murine typhus.

Nonetheless, although estimating the untreated mortality of murine typhus remains difficult, even when assuming a low fatality rate, murine typhus causes significant morbidity. Our review showed a mean duration of fever of up to 15 days and a high rate of hospitalisation and complications in untreated patients. Since treatment with doxycycline has been shown to lead to a cessation of fever with a mean of 3 days [4, 5] and a recent study [6] found significantly higher healthcare charges for murine typhus patients when compared with influenza, the importance of early diagnosis and treatment of this neglected disease cannot be overstated. That it is readily treatable with tetracyclines [5] should encourage more clinical awareness as an inexpensively and simply treatable infectious disease that can lead to complications.

## Supporting information

**S1 Checklist. PRISMA checklist.**
(DOC)

**S1 Fig.**
(DOCX)

**S1 Table.**
(DOCX)

**S2 Table.**
(DOCX)

**S3 Table.**
(DOCX)

## Acknowledgments

We would like to thank Professor Stephen Graves for his support in obtaining full-text articles, Professor Robin Bailey of LSHTM, the staff of the Lao-Oxford-Mahosot Hospital Wellcome Trust Research Unit (LOMWRU) and the Director and staff of the Microbiology Laboratory, Mahosot Hospital. We are very grateful to Bounthaphany Bounxouei, past Director of Mahosot Hospital, to Bounnack Saysanasongkham, Director of Department of Health Care, Ministry of Health and to H.E. Bounkong Syhavong, Minister of Health, Lao PDR.

## Author Contributions

**Conceptualization:** Paul N. Newton.

**Data curation:** Johannes F. Doppler.

**Formal analysis:** Johannes F. Doppler.

**Investigation:** Johannes F. Doppler.

**Methodology:** Johannes F. Doppler, Paul N. Newton.

**Project administration:** Johannes F. Doppler, Paul N. Newton.

**Resources:** Johannes F. Doppler, Paul N. Newton.

**Supervision:** Paul N. Newton.

**Validation:** Johannes F. Doppler.

**Visualization:** Johannes F. Doppler.

**Writing – original draft:** Johannes F. Doppler.

**Writing – review & editing:** Johannes F. Doppler, Paul N. Newton.

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
