## [Decision Letter · Decision Letter 0]

30 Mar 2020

Dear Dr. Doppler,

Thank you very much for submitting your manuscript "A systematic review of the untreated mortality of murine typhus" for consideration at PLOS Neglected Tropical Diseases. As with all papers reviewed by the journal, your manuscript was reviewed by members of the editorial board and by several independent reviewers. In light of the reviews (below this email), we would like to invite the resubmission of a significantly-revised version that takes into account the reviewers' comments. 

We cannot make any decision about publication until we have seen the revised manuscript and your response to the reviewers' comments. Your revised manuscript is also likely to be sent to reviewers for further evaluation.

Sincerely,

Job E Lopez, Ph.D.

Deputy Editor

Job Lopez

Deputy Editor

Reviewer's Responses to Questions

**Key Review Criteria Required for Acceptance?**

**Methods**

-Are the objectives of the study clearly articulated with a clear testable hypothesis stated?

-Is the study design appropriate to address the stated objectives?

-Is the population clearly described and appropriate for the hypothesis being tested?

-Is the sample size sufficient to ensure adequate power to address the hypothesis being tested?

-Were correct statistical analysis used to support conclusions?

-Are there concerns about ethical or regulatory requirements being met?

Reviewer #1: The applied methods are adequate and available in the manuscript and its supplements.

I would like to raise the following points to be either discussed or modified: 

1. A critical point is the reliable differentiation of R. typhi cases from R. prowazekii. The exclusion of studies only based on Weil-Felix is correct. Still the serological diagnosis of these two species is very difficult and requires techniques (e.g. western blot and cross-absorption) that are only available in few reference centers in the world (see Ref 7). Epidemiological criteria might be applied, since louse-borne typhus is not widely endemic, although, especially in older studies, there might be confusion with isolated Brill-Zinser cases. These problems are briefly mentioned in Conclusions, but could be placed with a bit more information in Discussion. I think, the CF test in the 40ie was an advance, but until today the serological separation of the 2 species remains a challenge. 

2. The study only found a very limited number of studies and patients. Therefore every included study/patient might add valuable information. The authors excluded studies with <10 untreated patients for selection bias. I would suggest to apply this to the total number of patients (treated + untreated). In a larger series of patients with few untreated cases, I do not see selection bias problems. This approach could help to include some of the 8 studies excluded for this reason.

3. Did the authors try to contact the respective author team of the newer publications to get some of the missing data?

Reviewer #2: (No Response)

**Results**

-Does the analysis presented match the analysis plan?

-Are the results clearly and completely presented?

-Are the figures (Tables, Images) of sufficient quality for clarity?

Reviewer #1: The results are clear and adequate.

One suggestion regarding Fig 2: due to the limited number of the included studies, the geographical "distribution" is rather random and not representative. The maps, although well prepared, are therefore of limited value and can be erased.

Reviewer #2: (No Response)

**Conclusions**

-Are the conclusions supported by the data presented?

-Are the limitations of analysis clearly described?

-Do the authors discuss how these data can be helpful to advance our understanding of the topic under study?

-Is public health relevance addressed?

Reviewer #1: L 248: Is there evidence from treated pediatric cases that confirm this hypothesis?

Discussion of diagnostic methods: see Methods

Reviewer #2: (No Response)

**Editorial and Data Presentation Modifications?**

Reviewer #1: (No Response)

Reviewer #2: (No Response)

**Summary and General Comments**

Reviewer #1: Well conducted study and well written manuscript.

Therefore, only minor comments and suggestions.

Reviewer #2: The manuscript by Doppler and Newton describe an analysis of the published literature to determine the untreated case fatality rate of those with murine typhus. In general, the manuscript is well written and attempts to clarify an important question. Below are specific comments for the authors’ consideration. 

Major comments: 

Line 86 – 89 and throughout: The authors aim to examine literature that excludes those with epidemic typhus (R. prowazekii) from those with murine typhus (R. typhi). The former has a much higher case fatality. Considering that an antibody response developed to one typhus group species is reactive to heterologous typhus group antigen, it is unclear how even serologic techniques, unless used with cross absorption, can differentiate between epidemic typhus and murine typhus. Thus, the method to exclude those with possible R. prowazekii infection is not entirely sound. Considering the low case fatality described here, I have no doubt that the cited articles are of those with murine typhus, but the limitation of even contemporary serologic assays, and the limitations as they apply here, should be more adequately discussed. 

Eligibility criteria: There is a wide variety of criteria used by various clinicians/investigators which are considered diagnostic. Can more detail be used to describe what was considered diagnostic (e.g., seroconversion, four-fold increase in titer, single antibody titers)? Although some of this detail is listed in table 1, it does not appear complete. 

Line 155: The majority of the patients in this series were hospital based. The case fatality from this study is 0.5%. In another study conducted by reviewing the hospital records of those with murine typhus (Dumler et al. JAMA. 1991;266:1365-70), 3.8% died despite treatment. Although the case series did not review cases in which hospital records where not available (many presumed by this reviewer to be outpatients with less severe disease), this hospital based mortality is quite different than what is reported here, despite the overwhelming majority receiving treatment. In another study reviewing cases over a 30-year period in Texas (a time period when effective antibiotics were available), the case fatality was reported to be 0.4% - close to what is reported here (Pieracci et al. AJTMH. 2017;96:1088-93). These differences and possible reasons for these differences in treated vs untreated case fatality should be discussed. 

Line 296 – 298: Although the highly cited review mentioned here may quote case fatalities of 1% and 4% in treated and untreated, respectively, other literature clearly states differently. As mentioned above, recommend discussing the case fatality rates described in other studies. 

Minor comments: 

Lines 77-82: The sentence here is a bit long and unwieldy. Suggest modifying to make more clear. 

Line 128: It is unclear what “patient characteristics” means here. Can the authors clarify in the manuscript? As written, it sounds almost like demographic data, which isn’t really a possible outcome. 

Lines 204 – 209: The publication by Crofton was said to be on the British Military. Could this have skewed the sex distribution? 

Lines 227 – 233: Much of the discussion is repetitious with other sections of the manuscript. Suggest streamlining repetitive material in favor of including more discussion to compare and contrast the case fatality here as that reported by others. 

Conclusions: Recommend consolidating the conclusion to a briefer form to emphasize key points. As written, it seems to almost be a rehash of the discussion section.

PLOS authors have the option to publish the peer review history of their article (what does this mean?). If published, this will include your full peer review and any attached files.

Reviewer #1: No

Reviewer #2: No
---

## [Decision Letter · Decision Letter 1]

26 Jul 2020

Dear Dr. Doppler,

We are pleased to inform you that your manuscript 'A systematic review of the untreated mortality of murine typhus' has been provisionally accepted for publication in PLOS Neglected Tropical Diseases.

Best regards,

Job E Lopez, Ph.D.

Deputy Editor

Job Lopez

Deputy Editor

Reviewer's Responses to Questions

**Key Review Criteria Required for Acceptance?**

**Methods**

-Are the objectives of the study clearly articulated with a clear testable hypothesis stated?

-Is the study design appropriate to address the stated objectives?

-Is the population clearly described and appropriate for the hypothesis being tested?

-Is the sample size sufficient to ensure adequate power to address the hypothesis being tested?

-Were correct statistical analysis used to support conclusions?

-Are there concerns about ethical or regulatory requirements being met?

Reviewer #1: (No Response)

Reviewer #2: Methods are sound.

**Results**

-Does the analysis presented match the analysis plan?

-Are the results clearly and completely presented?

-Are the figures (Tables, Images) of sufficient quality for clarity?

Reviewer #1: (No Response)

Reviewer #2: Results are presented clearly.

**Conclusions**

-Are the conclusions supported by the data presented?

-Are the limitations of analysis clearly described?

-Do the authors discuss how these data can be helpful to advance our understanding of the topic under study?

-Is public health relevance addressed?

Reviewer #1: (No Response)

Reviewer #2: Conclusions are supported and appropriate.

**Editorial and Data Presentation Modifications?**

Reviewer #1: (No Response)

Reviewer #2: (No Response)

**Summary and General Comments**

Reviewer #1: The authors have adequately responded to all questions. No further comments.

Reviewer #2: The authors have appropriately responded to this reviewer's questions, comments, and suggestions.

PLOS authors have the option to publish the peer review history of their article (what does this mean?). If published, this will include your full peer review and any attached files.

Reviewer #1: **Yes: **Thomas Weitzel

Reviewer #2: No

---

## [Editor Report · Acceptance letter]

4 Sep 2020

Dear Dr. Doppler,

We are delighted to inform you that your manuscript, "A systematic review of the untreated mortality of murine typhus," has been formally accepted for publication in PLOS Neglected Tropical Diseases.

Best regards,

Shaden Kamhawi

co-Editor-in-Chief

Paul Brindley

co-Editor-in-Chief
